# Primary Stability of Orthodontic Titanium Miniscrews due to Cortical Bone Density and Re-Insertion

**DOI:** 10.3390/ma13194433

**Published:** 2020-10-05

**Authors:** Gi-Tae Kim, Jie Jin, Utkarsh Mangal, Kee-Joon Lee, Kwang-Mahn Kim, Sung-Hwan Choi, Jae-Sung Kwon

**Affiliations:** 1Department and Research Institute of Dental Biomaterials and Bioengineering, Yonsei University College of Dentistry, Seoul 03722, Korea; gitae7673@yuhs.ac (G.-T.K.); kmkim@yuhs.ac (K.-M.K.); 2Department of Orthodontics, Institute of Craniofacial Deformity, Yonsei University College of Dentistry, Seoul 03722, Korea; kimj0515@yuhs.ac (J.J.); utkmangal@yuhs.ac (U.M.); orthojn@yuhs.ac (K.-J.L.); 3BK21 PLUS Project, Yonsei University College of Dentistry, Seoul 03722, Korea

**Keywords:** retrieved miniscrew, cortical bone density, primary stability, horizontal resistance, re-insertion, micromotion

## Abstract

The increasing demand for orthodontic treatment over recent years has led to a growing need for the retrieval and reuse of titanium-based miniscrews to reduce the cost of treatment, especially in patients with early treatment failure due to insufficient primary stability. This in vitro study aimed to evaluate differences in the primary stability between initially inserted and re-inserted miniscrews within different cortical bone densities. Artificial bone was used to simulate cortical bone of different densities, namely 20, 30, 40, and 50 pound per cubic foot (pcf), where primary stability was evaluated based on maximum insertion torque (MIT), maximum removal torque (MRT), horizontal resistance, and micromotion. Scanning electron microscopy was used to evaluate morphological changes in the retrieved miniscrews. The MIT, MRT, horizontal resistance, and micromotion was better in samples with higher cortical bone density, thereby indicating better primary stability (*P* < 0.05). Furthermore, a significant reduction of MIT, MRT, and horizontal resistance was observed during re-insertion compared with the initial insertion, especially in the higher density cortical bone groups. However, there was no significant change in micromotion. While higher cortical bone density led to better primary stability, it also caused more abrasion to the miniscrews, thereby decreasing the primary stability during re-insertion.

## 1. Introduction

Miniscrews are widely used as temporary anchorage devices in orthodontic treatments, thereby effectively preventing unwanted dentoalveolar side effects, as well as addressing poor patient cooperation and providing additional nonsurgical orthodontic treatment options [1]. However, the reported miniscrew failure rate varies from 0% to 40.8%, where the most recent meta-analyses have reported an early failure rate (within six months after implantation) of about 13.5% to 14% [2,3,4]. Early failure is mainly attributed to various factors related to weak integration between the miniscrew and bone, which leads to insufficient primary stability [4]. Bone quality is the main factor determining primary stability, while the insertion site characteristics, geometric design of miniscrew, operation technique, proximity to the root, periodontal inflammation, and loading time also play roles [4,5]. Insufficient primary stability leads to further orthodontic treatment, specifically re-insertion of the miniscrew [6,7].

Cortical bone quality, specifically thickness and density, plays the most critical role in primary stability. Some previous studies have reported that cortical bone thickness is the primary factor, as density does not affect primary stability when the thickness is sufficient (1.62 ± 0.57 mm in the maxilla and 2.13 ± 0.66 mm in the mandible), while others maintain that both factors are important [1,5,8].

The recent expansion in orthodontic treatment has led to a growing incidence of retrieved miniscrew re-insertion. This reduces the treatment cost, especially in cases of early failure due to insufficient primary stability [4,7]. Therefore, the performance of retrieved miniscrews should be further evaluated. Though some studies have focused on the insertion ability of retrieved miniscrews from clinical treatment or animal experiments, there are few studies about the primary stability of retrieved miniscrews, which is highly correlated with the success rate, especially in vitro studies with repeatability and standard condition to analyze influencing factors [6,7,9].

This study aimed to measure the primary stability of unused and retrieved miniscrews at different cortical bone densities and analyze morphological changes in the retrieved miniscrews to evaluate the mechanical properties that may influence the re-insertion process. The null hypothesis is that there is no difference in the primary stability of retrieved miniscrews compared with unused miniscrews in different cortical bone densities.

## 2. Materials and Methods 

### 2.1. Materials 

Orthodontic titanium-based miniscrews (OSSH1810, Osstem Implant, Busan, Korea) with a diameter of 1.8 mm and length of 10 mm were used (Figure 1). A polyurethane foam block (Sawbone, Division of Pacific Research Laboratories, Inc., Vashon, WA USA) was manufactured to simulate human bone according to a standard procedure, namely ASTM F-1839-08 (Table 1). The foam block density was defined as pounds per cubic foot (pcf), where 20, 30, 40, and 50 pcf foam blocks were produced to represent cancellous bone and cortical bone, as described in a previous study [8,10,11].

### 2.2. Sample Preparation

Artificial bone was produced according to a previously reported procedure and comprised two layers to simulate cortical and cancellous bone [9]. Specifically, 20 pcf foam (thickness = 20 mm) was used for cancellous bone, while 30, 40, and 50 pcf foam (thickness = 2 mm) were used for cortical bone (Figure 2). The two layers were bonded using a cyanoacrylate resin adhesive.

### 2.3. Torque Test

An automatic torque device (Admet eXpert8600, ADMET, Norwood, WA, USA) was used for the torque test, where the load-measurement and cross-head speed accuracy was ± 0.5%. The artificial bone was fixed with a clamp, and 5 rpm torque was applied clockwise using a torque wrench. Miniscrews (n= 6) were inserted into the artificial bone using a self-drilling system until only the head of the miniscrew was protruding, where a maximum load of 1.14 kgf was applied during the test. Miniscrew retrieval was performed using removal toque counterclockwise at 5 rpm with no applied load. Torque values were measured every second and computed using GaugeSafe software (ADMET, Norwood, WA, USA). The maximum insertion torque (MIT) and maximum removal torque (MRT) values were recorded. 

### 2.4. Horizontal Resistance

Horizontal resistance was evaluated according to a previously reported procedure [9]. The automatic torque device was used to insert unused miniscrews into artificial bone, where a universal testing machine (Instron 5942, Instron, Norwood, MA, USA) was used as the measuring device. A shear jig was used to hold the artificial bone as a knife-like device applied a tangential load to the head of the miniscrews at a crosshead speed of 1 mm/min (Figure 3). The applied force values and displacement data were computed using integrated software (Bluehill 2, Instron Corporation), where the force value at 0.6 mm displacement of the head was noted. 

### 2.5. Micromotion

Micromotion analysis was performed according to a previously reported procedure [12]. Immediately after miniscrew placement using the automatic torque device, micromotion was evaluated based on the Periotest value (PTV) (Medizintechnik Gulden, Modautal, Germany) and implant stability testing (IST) (AnyCheck, Neo Biotech, Seoul, Korea). The artificial bone was fixed using clamps to minimize movement and the instruments were calibrated before each measurement [13]. Each miniscrew was measured in triplicate to calculate a mean value. 

### 2.6. Morphological Evaluation

Scanning electron microscopy (SEM) (JEOL-7800F, JEOL Ltd. Tokyo, Japan) was performed at 15 kV acceleration voltage using uncoated samples, where images were acquired at 40× and 200× magnification according to a previously reported procedure [7]. Images of the thread edge and tip of four unused miniscrews were used as controls. Miniscrews were inserted into artificial bone with the various densities using the automatic torque device. The miniscrews were retrieved and cleaned as described in Section 2.4. SEM images of the retrieved miniscrews were acquired at the same positions for comparison with the unused miniscrews to evaluate the morphological changes. 

### 2.7. Evaluation of Re-Inserted Miniscrews

After completing the MIT, MRT and micromotion measurement, miniscrews were removed from the artificial bone using the automatic torque device. For the horizontal resistance test, retrieved miniscrews were prepared separately by automatic torque device. The retrieved miniscrews were ultrasonically cleaned in distilled water for 30 min to remove detritus around the miniscrew, and dried using an air gun [14]. The cleaned miniscrews were re-inserted into new artificial bone and measured again according to the same procedure.

### 2.8. Statistical Analysis

All statistical analyses were performed using IBM SPSS software (Version 26.0, IBM Korea Inc.), specifically one-way analysis of variance (ANOVA) with Tukey’s test and paired sample t-tests. A *P* value of less than 0.05 was considered statistically significant.

## 3. Results

### 3.1. Torque Test

The position and torque values were plotted against insertion time, as shown in Figure 4. The miniscrews consisted of three parts, namely a tip with a gradually increasing diameter, a middle portion with a fixed diameter, and an upper portion with the largest diameter (Figure 1). The change in torque value was divided into three stages based on this miniscrew design. The first stage involved the tip beginning to penetrate the cortical bone; thus, the implantation speed was slow. The torque increased rapidly as the diameter of the tip-portion increased. The second stage started as the tip had successfully penetrated the cortical bone. The implantation speed increased as the straight middle portion entered the cortical bone, leading either a slight increase, slight decrease, or no change in torque. The third stage involved the upper portion with the widest diameter entering the cortical bone, thereby causing a rapid increase in torque until the MIT was reached. 

The time required for the tip to penetrate the cortical bone increased with increasing cortical density. Behavior in the second stage was more complex, where the torque of the non-cortical bone group increased, while 30 pcf bone increased slowly, 40 pcf bone remained the same, and 50 pcf bone decreased. These results indicated that cortical bone damaged the miniscrews during insertion, which had an adverse effect that was more severe at higher cortical bone densities.

The MIT values of the initial insertion and re-insertion was 8.62 ± 0.72 and 8.28 ± 0.94 Ncm for non-cortical bone, 14.20 ± 0.99 and 11.07 ± 0.84 Ncm for 30 pcf bone, 18.80 ± 1.74 and 16.39 ± 1.14 Ncm for 40 pcf bone, and 24.58 ± 1.16 and 19.50 ± 1.81 Ncm for 50 pcf bone, respectively (Figure 5a). The MRT values of the initial insertion and re-insertion was 8.11 ± 0.69 and 6.85 ± 0.68 Ncm for non-cortical bone, 11.15 ± 0.39 and 9.82 ± 1.37 Ncm for 30 pcf bone, 15.92 ± 1.50 and 13.72 ± 1.06 Ncm for 40 pcf bone, and 22.91 ± 1.34 and 16.48 ± 4.33 Ncm for 50 pcf bone, respectively (Figure 5b). The MIT and MRT of the initial insertion and re-insertion increased with increasing cortical bone density. Furthermore, the MIT and MRT of the re-insertion exhibited different degrees of decline (*P* <0.05) within a given density.

The drop in MIT was more pronounced at higher cortical bone densities (Table 2). For example, the decrease in MRT between the non-cortical, 20 pcf, and 30 pcf bones exhibited a slight upward trend, but the difference was not significant. However, the decrease in MRT in the 50 pcf group was significantly higher than the other groups.

### 3.2. Horizontal Resistance

The horizontal force was plotted against deflection distance (Figure 6a), where the horizontal force was measured when deflection reached 0.6 mm (Figure 6b). The horizontal force of the initial insertion and re-insertion was 48.06 ± 5.81 and 45.59 ± 7.58 N for non-cortical bone, 93.22 ± 12.02 and 67.31 ± 8.11 N for 30 pcf bone, 111.31 ± 8.79 and 82.82 ± 17.53 N for 40 pcf bone, and 174.69 ± 7.65 and 133.59 ± 24.55 N for 50 pcf bone, respectively. Thus, horizontal force increased with increasing cortical bone density, where the force was lower for re-insertion compared to the initial insertion. 

The reduction of the horizontal force after re-insertion in 50 cpf bone was significantly higher than in non-cortical bone, and slightly higher than the 30 and 40 pcf bone, although the difference was not significant (Table 2).

### 3.3. Micromotion Test

PTV ranged between -8 and 50, where lower values indicate better stability [13]. An IST value below 59 indicated that the miniscrew was unstable, while between 60 and 64 indicated moderate stability and above 65 indicated high stability [13]. The PTV of the initial insertion and re-insertion was 2.44 ± 1.11 and 3.26 ± 0.72 for non-cortical bone, -0.84 ± 0.33 and -0.30 ± 0.75 for 30 pcf bone, -2.52 ± 0.44 and -2.18 ± 0.78 for 40 pcf bone, and -3.62 ± 0.41 and -4.12 ± 0.23 for 50 pcf bone, respectively (Figure 7a). The IST of the initial insertion and re-insertion was 59.2 ± 1.10 and 59.2 ± 0.84for non-cortical bone, 64.6 ± 1.34 and 62.2 ± 2.04 for 30 pcf bone, 68.6 ± 0.55 and 68.2 ± 1.30 for 40 pcf bone, and 71.4 ± 0.55 and 71 ± 1.22 for 30 pcf bone, respectively (Figure 7b). Overall, miniscrew micromotion decreased with increasing bone density. Neither PTV nor IST dropped significantly from the initial insertion to the re-insertion, and there was no significant difference between the reduction values at any density.

### 3.4. Morphological Evaluation

The SEM images visualized the sharp thread edges (Figure 8a) and narrow pointed tip (Figure 8i) of the control samples. In comparison, the retrieved miniscrews were worn down and blunt, especially after insertion in the high-density cortical bone (Figure 8h,p). 

## 4. Discussion

Cortical bone quality is one of the main factors influencing primary stability, and should thus be calculated [1,12,15]. Cortical bone density varies among different vertical skeletal facial profiles, as well as in different locations. Devlin et al. [16] reported that the densities of the maxilla anterior, maxilla posterior, and mandible are 0.55, 0.31 and 1.11 g/cm^3^, while another study reported that the average bone density of human mandible cortical bone is 0.664 g/cm^3^ [17]. Moon et al. [18] reported that the bone mineral density in the anterior was generally higher than in the posterior, where a decrease from the medial to lateral regions was observed. Therefore, an in vitro study of miniscrews under different cortical bone densities was required.

Human cadaver bones and fresh animal bones have been used as test materials in numerous previous studies [1,5,19,20,21]. However, both substrates lack homogeneity and consistency, and clear conclusions on influencing factors cannot be drawn. The American Society of Testing Materials (ASTM) [22] states that rigid polyurethane foam is an ideal material for testing miniscrews and other medical equipment. Specifically, the use of artificial bones as an alternative to human cadaver bones or fresh animal bones can avoid quality modifications over time and eliminate bias by controlling density and thickness, which are necessary for evaluating influencing factors [6,8].

Although several studies have reported that cortical bone density does not affect the primary stability when cortical bone thickness is sufficient (1.62 ± 0.57 mm in the maxilla and 2.13 ± 0.66 mm in the mandible), this study has demonstrated that both MIT and MRT increased with increasing of cortical bone density [5,23,24]. Some theories state that MIT and MRT are mainly determined by the tension-compression generated between the miniscrew and bone [25]. Friberg et al. [26] reported that a mandible with a higher density had higher torque than the maxilla, thereby supporting the findings of the current study. Furthermore, Motoyoshi et al. [3] reported that a torque of 5 to 10 Ncm was most conducive to maintaining the stability of miniscrews. With the exception of the non-cortical bone, all other sample groups in this study exceeded this range. Excessive torque may cause extreme cortical bone compression and micro-injury, which can affect stability and even cause necrosis in the nearby bone [27,28]. However, the study conducted by Motoyoshi was a clinical study, where many other variables in addition to torque affected the failure rate of miniscrews. Higher torque represents a tighter interface and more movement resistance in a standard in vitro study; thus, further study and insight into micro-injuries and bone necrosis are required [19].

Many biomechanical studies have proved that maximum insertion and removal torque may affect the primary stability of miniscrews, but this is not the only influencing factor [29]. Specifically, horizontal and pull-out resistance are also known to affect stability. However, miniscrews are mainly subjected to forces perpendicular to the long axis during orthodontic treatment; thus, a horizontal resistance test can provide more relevant data regarding primary stability compared to a pull-out test [11]. Moreover, most pull-out test results are far higher than typical maximum orthodontic forces in clinical treatment when miniscrews are not loosened [9]. Horizontal movement of 0.6 mm is considered a lack of miniscrew primary stability in clinical treatments, which can indicate a hidden risk of loss when subjected to orthodontic force [19]. Miniscrew displacement is a gradual process. Finite element analysis conducted by Dalstra [30] and Gallas et al. [31] demonstrated that a loading force perpendicular to the long axis of the miniscrew concentrates the maximum stress around the neck at the bone-implant interface. Simultaneously, horizontal traction produces a moment with the lower area of the cortical bone as the fulcrum due to the density difference between cortical bone and cancellous bone, thereby causing a tilting motion. Displacement of the miniscrew is dependent on elastic deformation of the miniscrew itself, as well as changes in the neck area of the cortical bone. Thus, horizontal resistance increased as the cortical bone density increased. A large traction force or extended traction can cause micro-damage of the cortical bone, which may lead to more compliance. This phenomenon requires further investigation [32].

Micromotion has become one of the most trusted methods for evaluating primary stability in recent years. Miniscrew micromotion profoundly affects bone regeneration [9], where a small degree of micromotion is vital for active bone reconstruction. However, even micro-scale micromotion can destroy new cells and blood vessels in the gap between the bone and miniscrew, causing osteoclasts to occupy the gap and absorb the bone tissue, resulting in decreased stability [33]. Larger micromotion should be regarded as a sign of implant failure. PTV and IST are two commercial hand-held micromotion measurement devices that obtain micromotion data by repeatedly tapping the implant [13]. Micromotion decreased with increasing cortical bone density in the current study, which was attributed to the higher porosity and elasticity of the low-density cortical bone. However, the accuracy of these easy-to-operate non-invasive devices for assessing implant stability remains a controversial topic [7,9,11,12]. 

Issues such as insufficient primary stability, incorrect implant position, gingival injury, and unwanted force direction during orthodontic treatment indicate that a miniscrew must be repositioned immediately [6,34]. The use of retrieved miniscrews from other patients has serious biological safety risks, but repeated use of a miniscrew in the same patient is not associated with the same risks and minimizes the patient’s medical expenses. Noorollahian et al. [35] reported that re-insertion of a miniscrew did not affect the insertion torque and removal torque, while Mattos et al. [36] found that torque was decreased. 

The MIT and MRT in the 50 pcf bone decreased significantly during re-insertion compared with the initial insertion, where the reduction was more significant than the other densities. The horizontal resistance in the 50 pcf bone also decreased significantly during re-insertion. SEM images (Figure 8) revealed that the thread edge of the retrieved miniscrews was worn-down and blunt, especially after use in high-density cortical bone. Migliorati et al. [37] reported that both MIT values and maximum load were mainly related to the depth of the thread and the thread shape factors. The changes of retrieved miniscrews in thread depth and shear area caused by abrasion and smoothness might lead to inadequate embedding properties and bone-miniscrew contact, adversely affect the torque and mechanical properties of the miniscrews.

Trisi et al. [38] reported that micromotion decreased as insertion torque increased, which was consistent with the findings of this study as the cortical bone density increased. However, re-insertion of the retrieved miniscrews did not cause an increase of micromotion. Thus, MIT is not necessarily representative of micromotion. Singh et al. [33] reported that the resistance of micromotion originates from the neck of the implant-bone interface. Self-drilling miniscrews were not affected by gaps formed during pre-drilling, indicating tight integration between the bone and both unused and retrieved miniscrews. Both PTV and IST measure micromotion by tapping the head of miniscrews horizontally; thus, the measured value is mostly determined by the hardness and elasticity of the cortical bone and miniscrew contact during tapping. This hardly changed in the retrieved miniscrews, unlike MIT, which is more affected by the morphological changes of the miniscrew (e.g., thread abrasion).

Primary stability is affected by many factors, thus primary stability cannot assessed based on a single measurement. However, this in vitro study clearly demonstrated performance degradation in retrieved miniscrews, especially regarding MIT, MRT, and horizontal resistance. Higher bone density facilitated higher primary stability, but caused more significant damage to the miniscrew during insertion and removal, resulting in an increased risk of failure during re-insertion. Therefore, the use of retrieved miniscrews in cases of high-density cortical bone is not recommended. However, limitations of in vitro study have to be taken into account; so far, the effect of roots, soft-tissue, orthodontic force, and jaw movement were not considered. Recently, there have been some studies focused on periodontal inflammation control or bone defects treatment during orthodontic treatment [39,40], suggesting that further research with different simulation design, especially about periodontal tissue and bone tissue changed caused by tooth movement, is necessary. Besides, while the artificial bone used in the present study was well manufactured for focusing on the effects and allowing us to investigate with standardized models where results would be consistent, it would not be identical to the natural bone in terms of chemical composition and physical integrity, especially the biological response to torque and thermal changes. Hence, the study may require further investigation with ex vivo or animal models to extrapolate to clinical practice. 

## 5. Conclusions

Higher cortical bone density led to better primary stability, but caused more damage to the miniscrews; this decreased primary stability during re-insertion. However, the artificial bone used in this study provided a limited simulation of the complex bone environment. Thus, further research is recommended.

## Figures and Tables

**Figure 1 materials-13-04433-f001:**
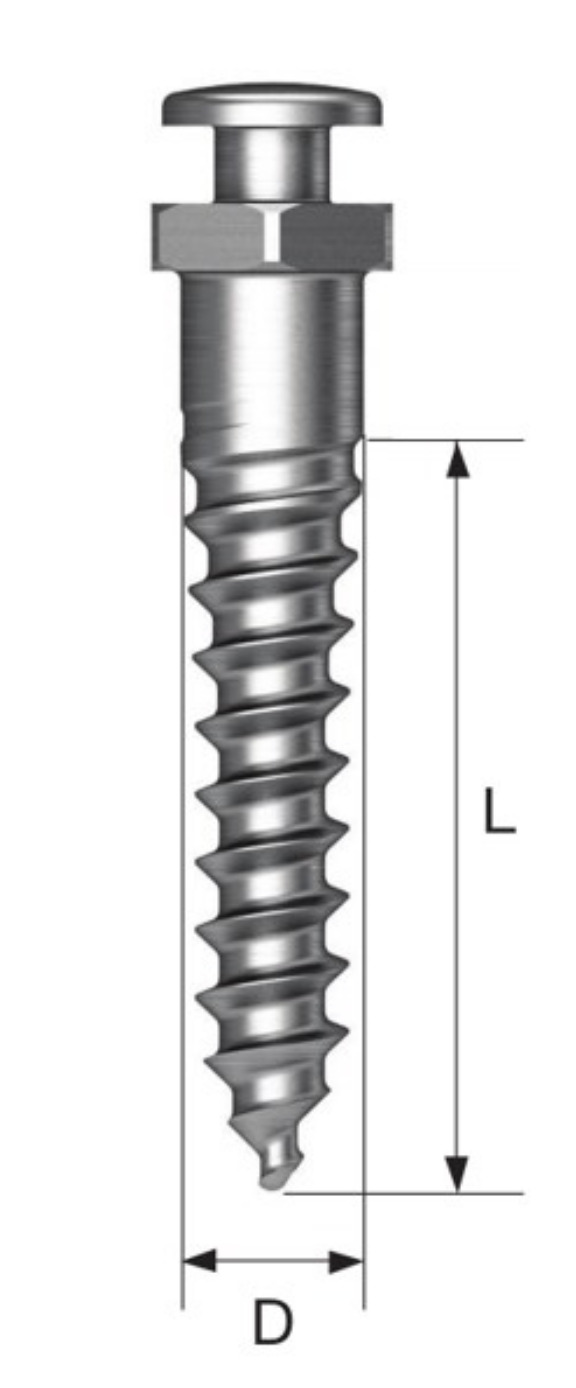
Schematic diagrams of the miniscrews (D—diameter, L—length, where D = 1.8 mm and L = 10 mm).

**Figure 2 materials-13-04433-f002:**
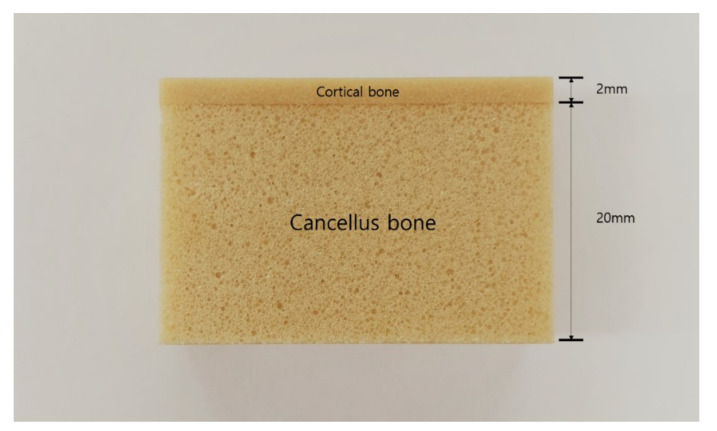
Digital photograph of a artificial bone block.

**Figure 3 materials-13-04433-f003:**
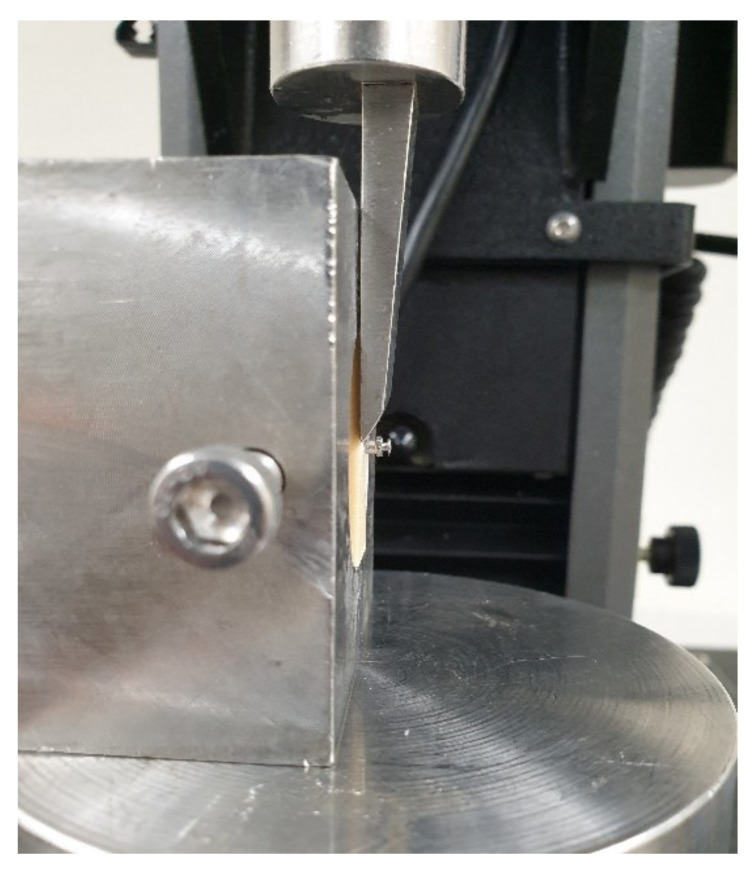
Digital photograph of the horizontal resistance test set-up.

**Figure 4 materials-13-04433-f004:**
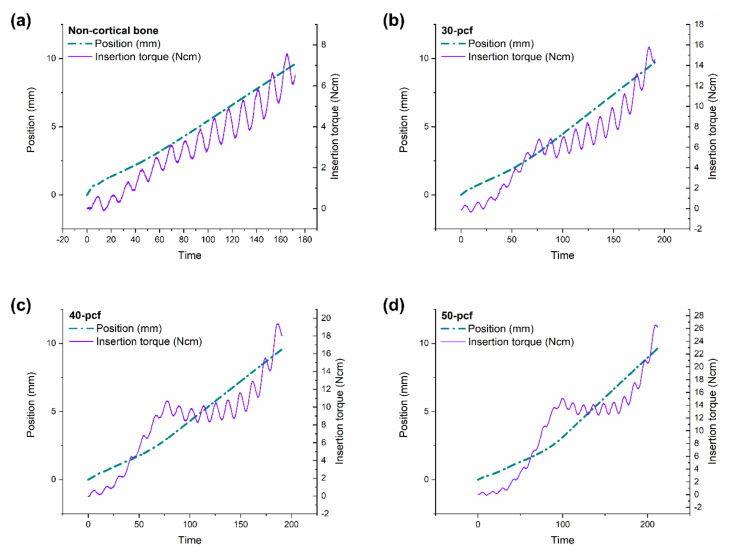
Position and torque values versus insertion time in (**a**) non-cortical bone, (**b**) 30, (**c**) 40, and (**d**) 50 pcf bone.

**Figure 5 materials-13-04433-f005:**
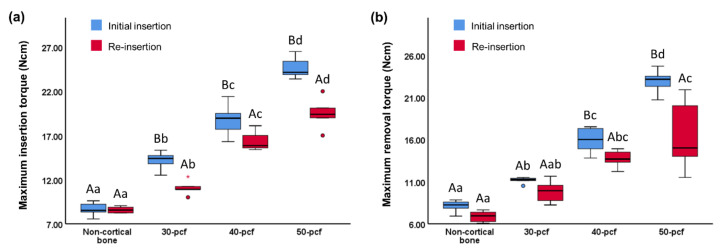
(**a**) Maximum insertion torque and (**b**) maximum removal torque values during the initial insertion and re-insertion of miniscrews into cortical bone with varying densities. Different uppercase letters indicate a significant difference between the initial insertion and re-insertion (*P* < 0.05). Different lowercase letters indicate a significant difference between the various cortical bone densities (*P* < 0.05).

**Figure 6 materials-13-04433-f006:**
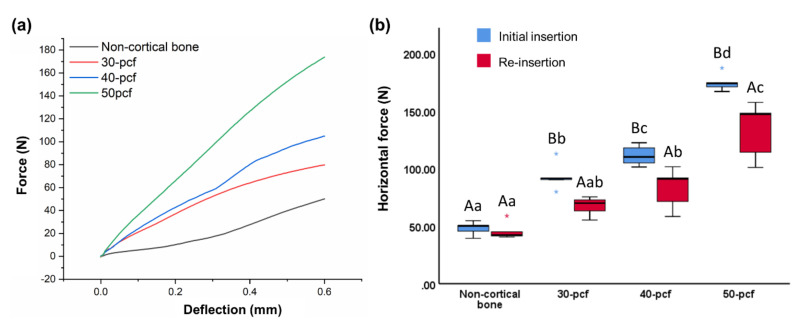
Horizontal resistance as (**a**) horizontal force versus deflection distance and (**b**) horizontal force at 0.6 mm deflection. Different uppercase letters indicate a significant difference between the initial insertion and re-insertion (*P* < 0.05). Different lowercase letters indicate a significant difference between the various cortical bone densities (*P* < 0.05).

**Figure 7 materials-13-04433-f007:**
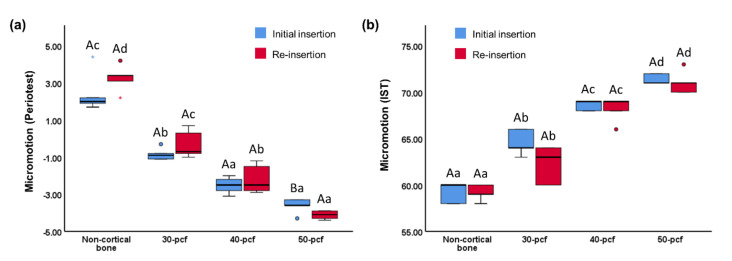
Micromotion values, namely (**a**) periotest value (**b**) implant stability testing value. Different uppercase letters indicate a significant difference between the initial insertion and re-insertion (*P* < 0.05). Different lowercase letters indicate a significant difference between the various cortical bone densities (*P* < 0.05).

**Figure 8 materials-13-04433-f008:**
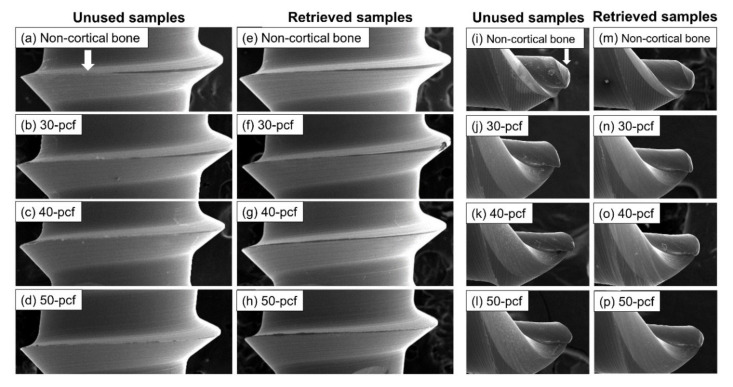
SEM images of miniscrew thread (40× magnification) and tip (200× magnification), namely (**a**–**d**) thread of unused samples, (**e**–**h**), thread of retrieved samples, (**i**–**l**) tip of unused samples, and (**m**–**p**) tip of retrieved samples.

**Table 1 materials-13-04433-t001:** Mechanical properties of the polyurethane foam bone blocks.

Density		Compression		Tension	
pcf	(g/cc)	Strength (MPa)	Modulus (MPa)	Strength (MPa)	Modulus (MPa)
**20**	0.32	8.4	210	5.6	284
**30**	0.48	18	445	12	592
**40**	0.64	31	759	19	1000
**50**	0.80	48	1148	27	1469

**Table 2 materials-13-04433-t002:** Reduction in primary stability between initial insertion and re-insertion of miniscrews into cortical bone with varying densities.

	Non-Cortical Bone Mean/SD	30 pcf Mean/SD	40 pcf Mean/SD	50 pcf Mean/SD
**MIT (Ncm)**	0.34 ± 0.45 ^a^	3.14 ± 0.93 ^b^	2.91 ± 0.80 ^b^	5.12 ± 0.79 ^c^
**MRT (Ncm)**	1.18 ± 0.66 ^a^	1.33 ± 1.58 ^a^	2.40 ± 0.86 ^a^	6.36 ± 3.28 ^b^
**Horizontal Resistance (N)**	2.47 ± 6.51 ^a^	25.91 ± 9.69 ^ab^	28.48 ± 17.19 ^ab^	41.10 ± 23.41 ^b^
**PTV**	−0.82 ± 1.21 ^a^	−0.54 ± 0.86 ^a^	−0.34 ± 1.09 ^a^	0.5 ± 0.55 ^a^
**IST**	0 ± 1.22 ^a^	2.4 ± 2.30 ^a^	± 1.67 ^a^	0.4 ± 1.34 ^a^

SD—standard deviation; Different lowercase letters indicate a significant difference between the various cortical bone densities (*P* < 0.05).

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
