# Peer review of "Primary Stability of Orthodontic Titanium Miniscrews due to Cortical Bone Density and Re-Insertion"

_materials, 2020, doi:10.3390/ma13194433_

Round 1
Reviewer 1 Report
Dear Authors
thanks for the sound study submitted. The study was well designed and well described.
However I would suggest to change the null hypothesis of the introduction section since it was previously stated that evidences are reported in the current literature supporting the effect of cortical bone density on primary stability.
Despite the soundness of every section of the manuscript, I think that in the Discussion you should analyze and describe the limitations of the invitro study.
This will for sure increase the readability and the clinical transfer of your research
Author Response
Q1
I would suggest to change the null hypothesis of the introduction section since it was previously stated that evidences are reported in the current literature supporting the effect of cortical bone density on primary stability.
A1
Thank you for the valuable suggestions. The null hypothesis was modified according to your comments as below;
This study aimed to measure the primary stability of unused and retrieved miniscrews at different cortical bone densities and analyze morphological changes in the retrieved miniscrews to evaluate the mechanical properties that may influence the re-insertion process. The null hypothesis is that there is no difference in the primary stability of retrieved miniscrews compared with unused miniscrews in different cortical bone densities.
Q2
Despite the soundness of every section of the manuscript, I think that in the Discussion you should analyze and describe the limitations of the in-vitro study. This will for sure increase the readability and the clinical transfer of your research
A2
Thank you and we agree with your comments. According to your suggestion, the analyze of limitations of the in-vitro study was added to the Discussion section as follows;
However, limitations of in-vitro study have to be taken into account; so far, the effect of roots, soft-tissue, orthodontic force and jaw movement were not considered. Recently, there have been some studies focused on periodontal inflammation control or bone defects treatment during orthodontic treatment [40,41], suggesting that further research with different simulation design, especially about periodontal tissue and bone tissue changed caused by tooth movement, is necessary.
We appreciate comments by reviewer and would like to thank for the time that you kindly offered to review this article and improved readability and the clinical transfer.
Reviewer 2 Report
- The manuscript evaluated the primary stability between initially inserted and re-inserted miniscrews within different cortical bone densities to guide whether the miniscrews can be reuse or not. This research has significance for cutting the patient’s medical expenses. All the experiments were well performed. However, the mechanism hidden behind the experiments data have not been further explain. Thus, the manuscript was not sufficiently sound for publication unless a minor revision is performed. The following suggestions should be considered:
- Please highlight the innovation of this article.
- Please explain mechanism of the difference regarding MIT, MRT, horizontal resistance between initially inserted and re-inserted miniscrews.
- The author repetitive described that the bone density would not affect the primary stable of the miniscrews when the bone possessed sufficient thickness. Please label the thickness.
- Please list some new ways for orthodontic and bone defects treatment. Some state-of-art works regarding repairing bone defects have to be included.(1) A strawberry-like Ag-decorated barium titanate enhances piezoelectric and antibacterial activities of polymer scaffold. Nano Energy, 2020, 74: 104825. (2)Low-level laser-aided orthodontic treatment of periodontally compromised patients: a randomised controlled trial. Lasers in Medical Science 35.3 (2020): 729-739.
Author Response
Q1
Please highlight the innovation of this article.
A1
Thank you for your thoughtful comments on the manuscript. According to your suggestion, the innovation of this study was added to the Introduction section.
Though some studies have focused on the insertion ability of retrieved miniscrews from clinical treatment or animal experiments, there are few studies about the primary stability of retrieved miniscrews, which is highly correlated with the success rate, especially in-vitro studies with repeatability and standard condition to analyze influencing factors [6-7, 9].
Q2
Please explain mechanism of the difference regarding MIT, MRT, horizontal resistance between initially inserted and re-inserted miniscrews.
A2
According to your suggestion, mechanism of the difference regarding MIT, MRT, horizontal resistance between initially inserted and re-inserted miniscrews was added to the Discussion section as follows:
Migliorati et al. [38] reported that both MIT values and maximum load were mainly related to the depth of the thread and the thread shape factors. The changes of retrieved miniscrews in thread depth and shear area caused by abrasion and smoothness might lead to inadequate embedding properties and bone-miniscrew contact, adversely affect the torque and mechanical properties of the miniscrews.
Q3
The author repetitive described that the bone density would not affect the primary stable of the miniscrews when the bone possessed sufficient thickness. Please label the thickness.
A3
According to your suggestion, the sentence was modified as follows:
Some previous studies have reported that cortical bone thickness is the primary factor, as density does not affect primary stability when the thickness is sufficient (1.62 ± 0.57 mm in the maxilla and 2.13 ± 0.66 mm in the mandible), while others maintain that both factors are important [1,5,8].
Although several studies have reported that cortical bone density does not affect the primary stability when cortical bone thickness is sufficient (1.62 ± 0.57 mm in the maxilla and 2.13 ± 0.66 mm in the mandible), this study has demonstrated that both MIT and MRT increased with increasing of cortical bone density.
Q4
Please list some new ways for orthodontic and bone defects treatment. Some state-of-art works regarding repairing bone defects have to be included.(1) A strawberry-like Ag-decorated barium titanate enhances piezoelectric and antibacterial activities of polymer scaffold. Nano Energy, 2020, 74: 104825. (2)Low-level laser-aided orthodontic treatment of periodontally compromised patients: a randomised controlled trial. Lasers in Medical Science 35.3 (2020): 729-739.
A4
According to your suggestion, some studies focused on periodontal inflammation control and bone defects treatment were added as references to the Discussion section.
Recently, there have been some studies focused on periodontal inflammation control or bone defects treatment during orthodontic treatment [40, 41], suggesting that further research with different simulation design, especially about periodontal tissue and bone tissue changed caused by tooth movement, is necessary.
Ref. 40. Ren, C.; McGrath, C.; Gu, M.; Jin, L.; Zhang, C.; Sum, F.; Wong, K.W.F.; Chau, A.C.M.; Yang, Y. Low-level laser-aided orthodontic treatment of periodontally compromised patients: a randomised controlled trial. Lasers Med Sci. 2020, 35, 729-739.
Ref. 41. Shuai, C.; Liu, G.; Yang, Y.; Qi, F.; Peng, S.; Yang, W.; He, C.; Wang, G.; Qian, G. A strawberry-like Ag-decorated barium titanate enhances piezoelectric and antibacterial activities of polymer scaffold. Nano Energy. 2020, 74.
Reviewer 3 Report
The article is well-written and present an interesting issue. However I have some suggestions:
Materials and methods
The reinsertion is not described in this section.
Were miniscrews tested also after reinsertion?
Results
Figures 5-7 should be presented as box and whisker plot.
Discussion
What are the limitations of the study?
Author Response
Q1
The reinsertion is not described in this section. Were miniscrews tested also after reinsertion?
A1
Thank you for your thoughtful comments and sorry for the limited information on the manuscript. We now have added sentence related to re-insertion to the Materials and method section.
2.7 Evaluation of re-inserted miniscrews
After completing the MIT, MRT and micromotion measurement, miniscrews were removed from the artificial bone using the automatic torque device. For the horizontal resistance test, retrieved miniscrews were prepared separately by automatic torque device. The retrieved miniscrews were ultrasonically cleaned in distilled water for 30 min to remove detritus around the miniscrew, and dried using an air gun [14]. The cleaned miniscrews were re-inserted into new artificial bone and measured again according to the same procedure.
Q2
Figures 5-7 should be presented as box and whisker plot.
A2
According to your suggestion, Figure 5, 6 and 7 are now presented as box and whisker plot in the Result section.
Q3
What are the limitations of the study?
A3
According to your suggestion, the analyze of limitations of the in-vitro study was added to the Discussion section as follows:
However, limitations of in-vitro study have to be taken into account; so far, the effect of roots, soft-tissue, orthodontic force and jaw movement were not considered. Recently, there have been some studies focused on periodontal inflammation control or bone defects treatment during orthodontic treatment [40,41], suggesting that further research with different simulation design, especially about periodontal tissue and bone tissue changed caused by tooth movement, is necessary.
Round 2
Reviewer 3 Report
Dear Authors,
Thank you for your changes.
Please add limitations associated with materials used (i.e. artificial bone) and methods used.
Author Response
Q1
Please add limitations associated with materials used (i.e. artificial bone) and methods used.
A1
Thank you for the valuable suggestions. The limitations associated with materials and methods was added to the Discussion section as follows:
Besides, while the artificial bone used in the present study was well manufactured for focusing on the effects and allowing us to investigate with standardized models where results would be consistent, it would not be identical to the natural bone in terms of chemical composition and physical integrity, especially the biological response to torque and thermal changes. Hence the study may require further investigation with ex-vivo or animal models to extrapolate to clinical practice.